# A Novel Approach to the Analysis of the Soil Consolidation Problem by Using Non-Classical Rheological Schemes

**Kazimierz Józefiak** *,† , **Artur Zbiciak** † , **Karol Brzeziński** † and **Maciej Maślakowski** †

Faculty of Civil Engineering, Warsaw University of Technology, 16 Armii Ludowej Ave., 00-637 Warsaw, Poland; a.zbiciak@il.pw.edu.pl (A.Z.); k.brzezinski@il.pw.edu.pl (K.B.); m.maslakowski@il.pw.edu.pl (M.M.)
* Correspondence: k.jozefiak@il.pw.edu.pl
† These authors contributed equally to this work.

**Abstract:** The paper presents classical and non-classical rheological schemes used to formulate constitutive models of the one-dimensional consolidation problem. The authors paid special attention to the secondary consolidation effects in organic soils as well as the soil over-consolidation phenomenon. The systems of partial differential equations were formulated for every model and solved numerically to obtain settlement curves. Selected numerical results were compared with standard oedometer laboratory test data carried out by the authors on organic soil samples. Additionally, plasticity phenomenon and non-classical rheological elements were included in order to take into account soil over-consolidation behaviour in the one-dimensional settlement model. A new way of formulating constitutive equations for the soil skeleton and predicting the relationship between the effective stress and strain or void ratio was presented. Rheological structures provide a flexible tool for creating complex constitutive relationships of soil.

**Keywords:** secondary consolidation; settlement; rheology; plasticity; constitutive modelling; soil mechanics; kepes



## 1. Introduction

Soil is a complex porous three-phase material in which many phenomena take place simultaneously. Because of the complexity a lot of soil mechanics problems have to be solved numerically or by laboratory investigation [1]. For numerical calculations, mainly using the finite element method (FEM), appropriate material models have to be implemented. As there are many kinds of soil types, differing in particle size fractions, geological origin and the conditions of loading, there is no universal solution. Many commercial FEM software packages provide their own material models [2]. However, they also allow for extending the models as there is often a need to analyse some less typical soil mechanics problems or soil types more thoroughly. In case of porous media, constitutive equations are generally formulated in the effective stress and applied for coupled pore pressure—stress numerical calculations of geotechnical problems [3].

Consolidation and settlement calculations are very important aspects of geotechnical design. The settlement of soil can be divided into three parts: (i) Elastic response, (ii) primary consolidation settlement related to the drainage of water, (iii) secondary settlement, related to the creep of the soil skeleton. In most cases only the primary consolidation is considered as it causes the largest settlement for the majority of soils. However, for organic or soft clay soils, the secondary compression can be also large and has to be taken into account [4]. Consolidation of the soft soil layer is also important in the case of column supported embankments, where the load transferred to the column and geosynthetic layer strongly depends on the settlement of the soft layer [5,6].

Rheological schemes provide a flexible tool for creating complicated constitutive relationships of various materials. By applying rheological schemes to model the behaviour of the soil skeleton and coupling them with the equation of pore water flow, it is possible

to describe the transient (time-dependant) response of soil taking into account secondary consolidation of viscoelastic soils [7–12]. In the study, a method of modelling the transient process of primary and secondary consolidation of soil based on Burgers rheological scheme is presented. The Burgers structure is commonly used to model the behaviour of asphalt mixtures [13,14]. Here, it will be applied for the soil skeleton in order to simulate the transient process of consolidation of an organic soil layer. It should be noted that the presented procedure can be applied for any viscoelastic rheological structures. This shows that the great potential of constitutive models developed in different research fields can be straightforwardly utilized to solve geotechnical problems. An example of such a problem is the modelling of secondary consolidation. The 1D constitutive equations derived using rheological schemes can be implemented as a part of the evolution law of the Cap yield surface of the Modified Drucker–Prager/Cap model available in commercial FEM codes.

On the other hand, the behaviour of soil in the effective stress space ($e - \log \sigma'$ curve) can be modelled by using non-classical rheological schemes as will be demonstrated in this paper. To do this, the non-classical Kepes element [15] will be introduced in a rheological scheme of the soil skeleton in effective stress, and based on that scheme a system of algebraic-differential equations will be formulated. This system of equations has not been presented previously in the geotechnical literature. Moreover, it should be emphasised that the Kepes element has not been applied to the geotechnical problem either. A detailed description of the Kepes rheological model for both 1D and 3D cases can be found in the monograph by Zbiciak [16].

## 2. One-Dimensional Consolidation Model

The one-dimensional consolidation problem is the consolidation of an infinite saturated homogeneous clay layer, loaded by an infinite and uniform pressure. The infinite layer of soil undergoing consolidation can be drained on both sides (see Figure 1) or only on one side and rests on a non-deformable base (e.g., rock). Only excess pore water pressure, which arises after application of the external load, is considered.

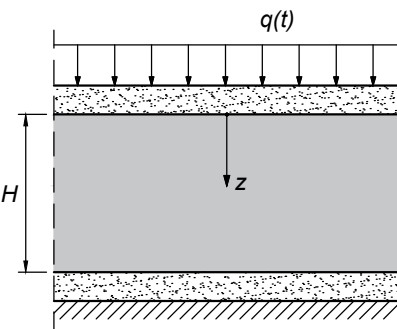

**Figure 1.** One-dimensional consolidation problem.

Despite the fact that one-dimensional formulation seems to be a simplistic approach it is often used for the prediction of the settlement of engineering structures, such as road embankments [17–19].

The simplest model of one-dimensional consolidation was presented by Karl Terzaghi [20]. The partial differential equation of flow can be expressed as:

$$\frac{k}{\gamma_w} \frac{\partial^2 \sigma'(z,t)}{\partial z^2} = \frac{\partial \varepsilon(z,t)}{\partial t} \tag{1}$$

where $k$ is the coefficient of permeability, $\gamma_w$ is the unit weight of water, $\sigma'$ is the effective stress function of depth and time, and $\varepsilon$ is strain function of depth and time.

Assuming the linear constitutive law for the soil's skeleton, in which the strain function $\varepsilon(z, t)$ is related to the effective stress $\sigma'(z, t)$ by the following equation

$$\varepsilon(z, t) = \frac{\sigma'(z, t)}{M} \tag{2}$$

where $M$ denotes the oedometer modulus, Equation (1) can be solved with appropriate initial and boundary conditions. For the two-side drainage (see Figure 1) the boundary conditions are as follows

$$\sigma'(0, t) = q(t) \qquad \sigma'(H, t) = q(t) \tag{3}$$

where $q(t)$ is a function of the applied load and $H$ is the thickness of the soil layer. The initial condition is $\sigma'(z, 0) = 0$. Equation (1) can be solved both analytically and numerically [20]. In the paper [10] numerical solutions of the Terzaghi problem were demonstrated. Moreover, some more advanced rheological schemes modelling the soil skeleton were also taken into consideration and the results of numerical solutions were compared with known analytical formulas [8].

The simple linear relationship between the effective stress and strain (Equation (2)) does not take into account secondary consolidation to be observed in organic or soft clay soils. This can be shown by comparison of the model prediction with oedometer laboratory test data carried out on a remoulded sample of soil with about 7% organic matter content (see Figure 2). Taking different values of parameters $M$ and $k$, it was impossible to obtain a satisfying fit to the oedometer test results.

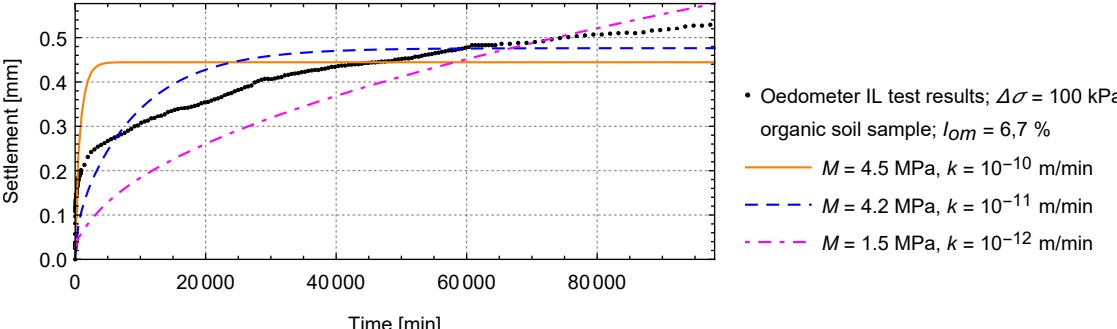

**Figure 2.** Terzaghi solution in comparison to test data.

## 3. Classical Rheological Schemes for Consolidation

Various classical rheological structures, represented by connections of springs and dashpots, are widely used to model the rate-dependent behaviour of asphaltic materials (see [21–25]) and polymers [26]. Such schemes for modelling problems of soil mechanics were adopted, e.g., in papers [7,8]. Gibson and Lo [8] presented an analytical solution of one-dimensional consolidation in which the relationship between the effective stress and vertical strain was formulated according to the standard model.

### 3.1. Burgers Model for Soil Skeleton

In this section, we introduce another rheological scheme not widely used in soil mechanics—the Burgers model (see Figure 3). This model is commonly used to model the behaviour of asphalt mixtures [13,14]. Here, it will be applied for the soil skeleton to simulate the transient soil response during consolidation and to obtain the settlement change in time.

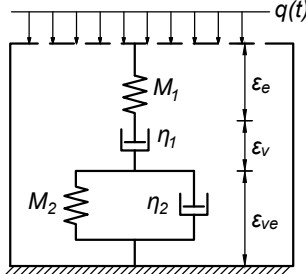

**Figure 3.** Burgers model applied for the soil skeleton for transient consolidation modelling.

Assuming the Burgers model for the soil skeleton (see Figure 3), the constitutive relationship can be formulated as follows [10]:

$$\eta_2 M_1 \dot{\varepsilon} + M_1 M_2 \varepsilon = \eta_2 \dot{\sigma}' + M_1 M_2 \varepsilon_v \tag{4}$$

where the strain function $\varepsilon_v(z, t)$ can be determined by solving the following differential equation

$$\dot{\varepsilon}_v = \frac{1}{\eta_1} \sigma' \qquad \varepsilon_v(z, 0) = 0 \tag{5}$$

and the superimposed dot denotes the derivative with respect to time. In Equations (4) and (5) the rheological structure's parameters (see Figure 3) $M_1$ and $M_2$ can be interpreted as the oedometric (constrained) moduli and $\eta_1$ and $\eta_2$ are the viscosity constants.

Equations (4) and (5) are an extension of the linear relationship (2). Thus, in order to simulate the process of transient consolidation, the system of coupled differential equations (1), (4) and (5) has to be solved with the boundary conditions (3) and 'zero' initial conditions for $\varepsilon(z, t)$ and $\varepsilon_v(z, t)$. After solving the initial boundary value problem, the function of displacement (settlement) can be found by integrating the strain function.

Advanced elastic-plastic models implemented in various geotechnical finite element method software packages can be extended by taking into account soil's skeleton creep. For example the Drucker–Prager/Cap model available in Abaqus softwere [27] gives the possibility to implement the constitutive relationship between the creep volumetric strain and effective pressure (first invariant of the effective stress tensor). Thus, the one-dimensional considerations shown in this section can be applied to the existing three-dimensional models. In order to implement such extension, in Equations (4) and (5), 1D strain, $\varepsilon$, should be replaced by volumetric strain, $\varepsilon_{vol} = \text{tr}\,\boldsymbol{\varepsilon}$, and $\sigma'$ by effective pressure $p' = -\frac{1}{3}\text{tr}\,\boldsymbol{\sigma}'$.

### 3.2. Numerical Solution

The system of partial differential equations was solved using the numerical method of lines implemented in Mathematica software [10,28]. The assumed parameters of the Burgers model are shown in Table 1. The comparison with the example test data is shown in Figure 4. Comparing Figures 2 and 4 shows that, with this approach, the oedometer test results can be predicted a lot better than they were in the case of the simple elastic relationship (2). In order to achieve even better fit of the model, the parameters of the model can be found by an automatic curve fitting to the laboratory test data. The procedure and results will be shown in future works.

**Table 1.** Parameters assumed for the Burgers model in transient analysis for comparison with test data.

| $M_1 \, (\text{MPa})$ | $M_2 \, (\text{MPa})$ | $\eta_1 \, (\text{MPa} \cdot \text{min})$ | $\eta_2 \, (\text{MPa} \cdot \text{min})$ | $k \, (\text{m/min})$ |
|---|---|---|---|---|
| 14.0 | 10.0 | $1.2 \times 10^6$ | $45 \times 10^3$ | $1 \times 10^{-8}$ |

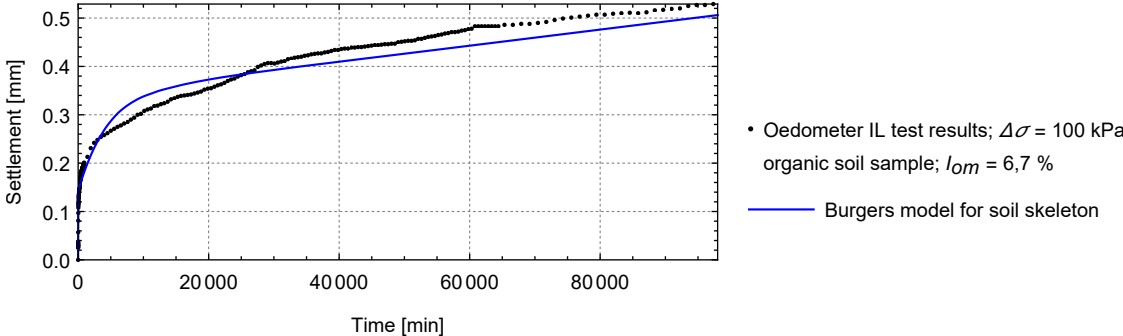

**Figure 4.** Comparison of the solution with Burgers model with long-term incremental loading (IL) oedometer test results.

## 4. Kepes Element for Effective Stress Behaviour

In Section 3 transient response of soil was analysed, that is the change of settlement in time related to the drainage of pore water and rheological effects in the soil skeleton. Models of transient consolidation can be used to predict settlement of a structure after a certain amount of time, for example, in order to calculate how much settlement occurs during the time of construction. In this section models for different conditions will be presented. Here, we introduce a non-classical rheological element to model the behaviour of soil in effective stress—that is only the response of the soil skeleton in drained conditions. Such models of steady-state consolidation can be used for calculation of long-term (maximum) settlement which occurs when there is no excess pore pressure.

### 4.1. Formulation

Kepes model describes materials which dissipate energy. It is a special case of a plastic material, in which the set of the admissible stress depends on the strain state. The process of energy dissipation for the Kepes material as well as the plastic material is rate independent. The phenomenon of rate independency makes Kepes and plastic elements different from the classic viscous element (dashpot). The assumption is made that for an undeformed body stresses are zero. The one-dimensional constitutive relationship for the Kepes model is similar to the relationship of a classical slider element which describes ideal plasticity but with the plastic limit not constant but dependent on the strain: $\sigma_0(\varepsilon)$. For the one-dimensional case the continuous set of admissible stress for the Kepes material is as follows [16]

$$\Theta_{kp} := [-\sigma_0; \sigma_0] \tag{6}$$

where $\sigma_0 = \kappa|\varepsilon|$ and $\kappa$ is a material constant. The Kepes element can be effectively used to model the behaviour of soil in effective stress as observed in oedometer or isotropic consolidation tests. The typical $e - \log \sigma'$ curve obtained from these tests is shown in Figure 5. While we are going to formulate the relationship between vertical strain (equivalent to the volumetric strain for the assumed conditions) and stress, the void ratio $e$ can be calculated, for a given soil's initial void ratio $e_0$, using the well-known formula:

$$e = \varepsilon(1 + e_0) + e_0 \tag{7}$$

The first model we will analyse is a connection in series of the Kepes element and a spring (see Figure 6a). To formulate the constitutive relationship in effective stress for such a scheme, let us first denote the logarithm of the effective stress as

$$\sigma'_L := \begin{cases} -\log|\sigma'| & \text{for} \quad \sigma' \leq 0 \\ \log|\sigma'| & \text{for} \quad \sigma' > 0 \end{cases} \tag{8}$$

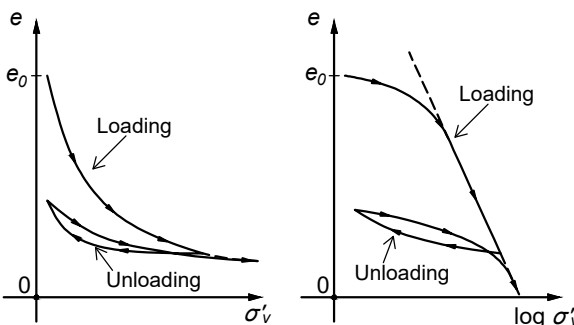

**Figure 5.** Typical $e - \log \sigma'$ relationship.

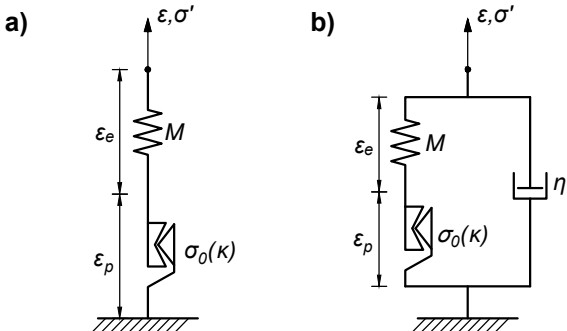

**Figure 6.** Rheological schemes proposed for drained conditions: (**a**) Spring and Kepes element connected in series, (**b**) Spring and Kepes element connected in parallel with dashpot.

In the definition (8) natural logarithm can also be used. The choice depends on the way results of oedometer or isotropic consolidation tests are presented. Here, as the common logarithm is used, the slopes of the virgin compression line and the unloading–reloading line will be related to the compression index ($C_c$) and to the swelling index ($C_s$), respectively.

Now, for the Kepes element we can define $\sigma'_L = \beta\sigma'_0$ where $\sigma'_0 = \kappa|\varepsilon_p|$, so:

$$\sigma'_L = \beta\kappa|\varepsilon_p| \tag{9}$$

and the derivative with respect to time is:

$$\dot{\sigma}'_L = \dot{\beta}\kappa|\varepsilon_p| + \beta\kappa\frac{\varepsilon_p}{|\varepsilon_p|}\dot{\varepsilon}_p \tag{10}$$

For elements connected in a series, as shown in Figure 6a, the stress in the spring is

$$\sigma'_L = M(\varepsilon - \varepsilon_p) \tag{11}$$

where $M$ is the oedometric (constrained) modulus. Thus, the strain in the Kepes element is

$$\varepsilon_p = \frac{1}{M}(M\varepsilon - \sigma'_L) \tag{12}$$

hence

$$\dot{\varepsilon}_p = \dot{\varepsilon} - \frac{\dot{\sigma}'_L}{M} \tag{13}$$

Substituting Equation (10) to Equation (13) we get

$$\dot{\varepsilon}_p = \dot{\varepsilon} - \frac{1}{M}\left(\dot{\beta}\kappa|\varepsilon_p| + \beta\kappa\frac{\varepsilon_p}{|\varepsilon_p|}\dot{\varepsilon}_p\right) \tag{14}$$

After some algebra, Equation (14) can be rewritten as

$$\dot{\varepsilon}_p = \frac{|\varepsilon_p|\left(M\dot{\varepsilon} - \dot{\beta}\kappa|\varepsilon_p|\right)}{M|\varepsilon_p| + \beta\kappa\varepsilon_p} \tag{15}$$

where

$$\beta = \frac{\sigma'_L}{\sigma'_0} = \frac{M(\varepsilon - \varepsilon_p)}{\kappa|\varepsilon_p|} \tag{16}$$

It can be shown that assuming only compression and no tension in the model (which is the case for soils), Equation (15) can be reduced to the following form [16]:

$$\dot{\varepsilon}_p = \left[\frac{|\varepsilon_p|M\dot{\varepsilon}}{M|\varepsilon_p| - \kappa\varepsilon_p}\right]^{-} \tag{17}$$

where the projection onto the set of negative numbers was used:

$$[z]^{-} := \begin{cases} z & \text{if} \quad z < 0 \\ 0 & \text{if} \quad z \geq 0 \end{cases} \tag{18}$$

Numerical implementation needs the case of $\varepsilon_p = 0$ to be treated in a special manner to avoid indeterminate formulas (see Equation (17)). Only for this case can an appropriate spring element be substituted instead of the Kepes element, so

$$\dot{\sigma}'_L = M(\dot{\varepsilon} - \dot{\varepsilon}_p) \tag{19}$$

where $\dot{\sigma}'_L = \kappa\dot{\varepsilon}_p$. Hence

$$\dot{\varepsilon}_p = \frac{M}{\kappa + M}\dot{\varepsilon} \quad \text{for} \quad \varepsilon_p = 0 \tag{20}$$

Equation (17) can be solved for a given strain excitation function $\varepsilon(t)$. After determining $\varepsilon_p(t)$, the logarithmic stress can be calculated using Equation (11). Then, linear effective stress $\sigma' = 10^{\sigma'_L}$ (compare Equation (8)).

The model presented in Figure 6a is rate independent, as was stated in the introduction of Section 4. Thus, the results of simulation will not depend on the velocity of applied strain excitation. In order to take into consideration the effect of rate dependency, the scheme presented in Figure 6a can be modified by adding an additional element, for example, a dashpot. This way, secondary consolidation phenomena can be taken into account.

The extended model is presented in Figure 6b. After introducing a dashpot connected in parallel, Equation (11) becomes:

$$\sigma'_L = M(\varepsilon - \varepsilon_p) + \eta\dot{\varepsilon} \tag{21}$$

where $\eta$ is viscosity. For the viscosity parameter $\eta = 0$ model presented in Figure 6a is obtained.

*4.2. Numerical Tests*

The equations were tested for the following function of strain

$$\varepsilon(t) = At\sin(\omega t) + Bt \tag{22}$$

in which $A = 3 \times 10^{-7}\frac{1}{\text{min}}$ and $\omega = 0.025\frac{1}{\text{min}}$ were assumed. In order to test the model's prediction for different strain rates, two values of the parameter $B$ were assumed: $B = 1 \times 10^{-5}\frac{1}{\text{min}}$ and $B = 1 \times 10^{-6}\frac{1}{\text{min}}$. The functions of the strain excitation are shown in Figure 7. The model was tested for parameters shown in Table 2. The idea was to find a function which can approximate a consolidation test and use it to verify the applicability of the model.

**Table 2.** Parameters assumed for numerical tests of the schemes including the Kepes element.

| $M[\textbf{MPa}]$ | $\kappa[\textbf{MPa}]$ | $\eta[\textbf{MPa} \cdot \textbf{min}]$ |
|---|---|---|
| 6.5 | 4.0 | 0 and 20 |

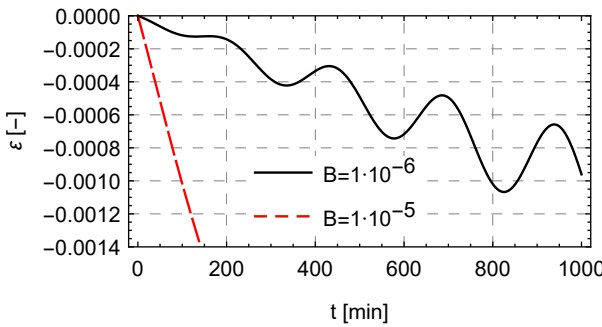

**Figure 7.** Strain excitation functions applied.

Figure 8 shows the relation between $\log \sigma'$ and the vertical strain obtained by the model for $\eta = 0$, that is for the rate-independent scheme (see Figure 6a). The produced curve is the same as the idealised graph obtained in consolidation laboratory tests. The relation between vertical effective stress and strain is shown in Figure 9. Corresponding curves after adding rate-dependency, by setting the viscosity parameter $\eta = 20\,\text{MPa} \cdot \text{min}$, are shown in Figures 10 and 11. In Figure 10 one can see that the normal consolidation line shifts to the right-hand side for the lower strain rate. This phenomenon is confirmed by many researchers for soil exhibiting secondary compression [29].

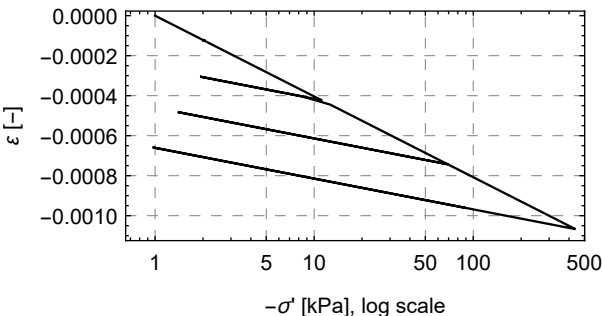

**Figure 8.** Relationship between $\log \sigma'$ and strain produced by the model for $\eta = 0$.

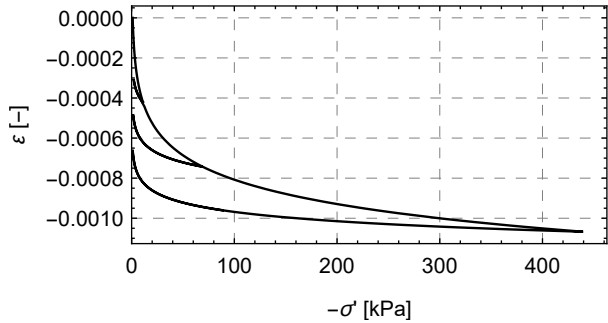

**Figure 9.** Relationship between $\sigma'$ and strain produced by the model for $\eta = 0$.

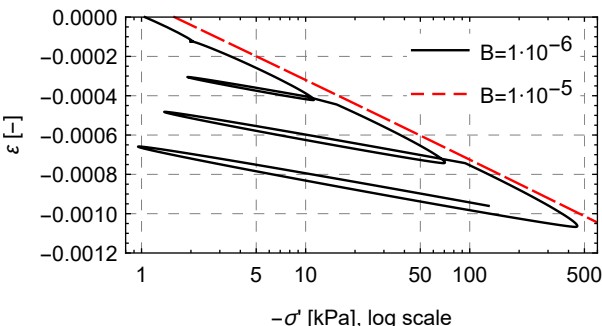

**Figure 10.** Relationship between $\log \sigma'$ and strain produced by the model for $\eta = 20$ MPa $\cdot$ min for two different strain rates.

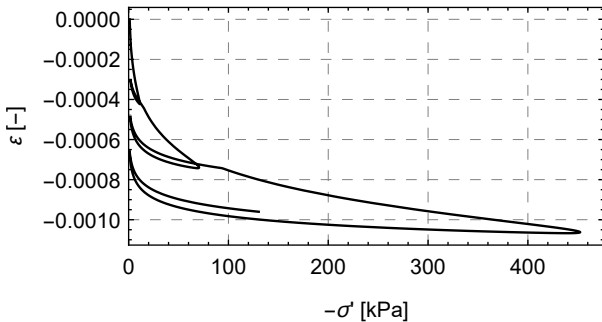

**Figure 11.** Relationship between $\sigma'$ and strain produced by the model for $\eta = 20$ MPa $\cdot$ min.

In order to take into account pre-consolidated soils, a simple modification of Equation (16) should be done by introducing an additional term in the denominator as follows:

$$\beta = \frac{\sigma'_L}{\sigma'_0} = \frac{M(\varepsilon - \varepsilon_p)}{\kappa |\varepsilon_p| + \log |\sigma'_p|} \tag{23}$$

where $\sigma'_p$ is the preconsolidation pressure. The simulation results of the preconsolidated soil model, assuming $\sigma'_p = 50$ kPa, are visualized in Figure 12.

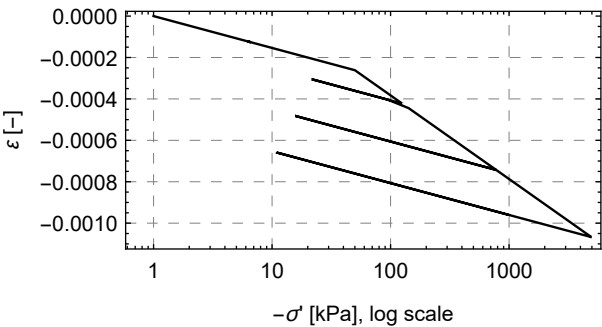

**Figure 12.** Relationship between $\log \sigma'$ and strain produced by the preconsolidation model for $\eta = 0$ and $\sigma'_p = 50$ kPa.

## 5. Discussion

FEM software with an open structure like, for example, Abaqus gives the possibility to implement parts of constitutive models (e.g., evolution of the yield surface) by applying user-defined subroutines. In particular, in case of the Modified Drucker–Prager/Cap model, the evolution of the Cap yield surface is defined in terms of the volumetric inelastic strain, that being the sum of plastic volumetric strain and creep volumetric strain ($\varepsilon^{cr}_{vol}$). The $\varepsilon^{cr}_{vol}$ is related to modelling the secondary consolidation phenomenon. The consolidation creep mechanism in the Modified Drucker–Prager/Cap is defined by specifying a constitutive

law for $\varepsilon_{vol}^{cr}$ in a form of differential equation in terms of an equivalent creep pressure and $\varepsilon_{vol}^{cr}$. Three laws for creep are implemented in Abaqus, one of them being the Singh–Mitchell law. However, any equation for calculating $\varepsilon_{vol}^{cr}$ can be given by a user subroutine CREEP. Thus, the equations presented in the manuscript for transient consolidation can be implemented as an extension to the Modified Drucker–Prager/Cap model. Rheological structures serve as a useful tool for building creep laws. In the article, the Burgers structure was analysed in detail; however, even more complicated viscoelastic models can be used in order to relatively easily find equations describing creep.

The steady-state consolidation model presented in the manuscript, in which the Kepes element was used, can be also utilized in order to build and to implement a complete material model for consolidation analysis. Geotechnical FEM software (i.e., Plaxis) allows user definitions of material models programmed in Fortran. The authors plan to extend the presented approach to a three-dimensional case and implement the constitutive model using the UMAT user subroutine available in Abaqus software.

The right-hand branch of the steady-state consolidation model can be further modified to obtain a more suitable solution. This will be done in future works. Moreover, both incremental loading (IL) and constant rate of strain (CRS) oedometer tests will be carried out by the authors in order to further validate the proposed model.

The equation of flow can be modified to simulate the flow of water more realistically [4]. In a transient FEM analysis the equations of flow are solved numerically. Moreover, according to test data, the coefficient of permeability changes in time as the consolidation progresses and the density of the material increases [30]. The authors plan to assume a non-constant function of this parameter in the equation of flow in future works.

## 6. Conclusions

Transient one-dimensional consolidation of soil was modelled by using the Burgers rheological scheme and numerical integration of differential equations. Comparison with laboratory test data shows that the model provides good prediction of the soil's behaviour and an iterative curve fitting procedure can be implemented for finding the model's parameters based on laboratory tests. Constitutive equations of the soil skeleton, formulated here for the one-dimensional transient consolidation case, can be also applied for the isotropic consolidation conditions.

Additionally, a new way of formulating constitutive equations for the soil skeleton and predicting the relationship between the effective stress and strain in drained conditions ($e - \log \sigma'$ curve) was presented. The set of equations for the non-classical Kepes rheological element connected in series with a spring was presented. The model was then extended by a simple addition of a rate-dependent dashpot element which allowed to take into account secondary consolidation effects. Testing the model for two different strain rates showed the well-known phenomena of the normal consolidation line shift. The presented solution can be treated as an extension or alternative to the Isotache model being developed by many researchers [29,31].

The main findings of this paper can be summarized in the following points:

- A method of modelling the transient process of primary and secondary consolidation of soil based on rheological schemes was presented. In the paper Burgers structure was used as an example; however, the presented procedure can be applied for any viscoelastic rhelogical structures. The 1D constitutive equations derived using rheological schemes can be implemented as part of the evolution law of the Cap yield surface in commercial finite element method codes.
- A new way of formulating constitutive equations for the soil skeleton in steady-state (drained) conditions using the non-classical Kepes element was presented.
- An explicit form of differential equations was formulated for the non-classical Kepes rheological element.
- Presented rheological models are able to describe the material response for cyclic loading conditions: Both for loading and unloading.

- Formulated equations can be solved using known algorithms for solving ordinary differential equations (e.g., the Runge–Kutta method).

**Author Contributions:** Conceptualization, K.J., A.Z., M.M., K.B.; methodology, K.J. and A.Z.; software, K.J.; validation, A.Z., KB.,M.M.; formal analysis, A.Z.; resources, K.J., M.M.; data curation, K.J.; writing–original draft preparation, K.J.; writing–review and editing, A.Z., K.B.; visualization, K.J.; supervision, A.Z. All authors have read and agreed to the published version of the manuscript.

**Funding:** This research received no external funding.

**Institutional Review Board Statement:** This study did not require ethical approval.

**Informed Consent Statement:** Not applicable.

**Data Availability Statement:** The data supporting the findings of this study are available within the article.

**Conflicts of Interest:** The authors declare no conflict of interest.

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
