# Peer review of "A Novel Approach to the Analysis of the Soil Consolidation Problem by Using Non-Classical Rheological Schemes"

_applsci, doi:10.3390/app11051980_

Round 1
Reviewer 1 Report
The manuscript is well-written, and the research presented is interesting. The numerical analysis fit well experimental data, and therefore, the validity of the numerical model was properly accomplished by the authors. However, I have some concerns about the novelty and impact of this study that should be clarified:
- First, the introduction needs to clearly identify the novelty of the study and how this model will fill this gap of knowledge.
- Second, although the abstract is very well written and structured, the last sentence is not properly supported by the results and conclusions and also reflect the previous point. The authors state “These relationships can be implemented within various kinds of finite element method (FEM) software as an extension to classical material models.” Therefore, the authors need to compare the results of their work with the results of models already implemented in FEM software. Basically, a form of benchmarking.
Minor issues:
- Units in Table 1 should be consistent thought the manuscript: MPa.min or .s
- Same for plots, e.g. Figures 7 and 8 Strain vs ε
Reviewer 2 Report
This paper addresses the problem of secondary consolidation in the soils with high organic content using less-common rheological schemes. The paper has a solid theoretical approach, and can be considered for publication. Here are few comments.
- Lines 63-64; the initial condition of the consolidation equation in terms of effective stress should indicate that the additional effective stress is zero for the entire depth of clay layer. Alternatively, if the consolidation equation is written in terms of pore water pressure, the initial condition should indicate that the excess pore water pressure for the entire depth of clay layer is equal to the surcharge, which is q. Please revise the initial condition accordingly.
- Please make sure that all of the parameters used in the equations are defined immediately after the equation.
- Lines 89-90; Does the systems of differential equation for the proposed rheological model shown in Figure 3 include the equations (1), (4) and (5), or just (4) and (5)? Please explain.
- Line 149; is Eq. (15) being reduced to (17)?
- I am not sure if the paper clearly addresses the case of over-consolidated soils as noted in the abstract.
- The second part of this paper (Section 4) seems to be about primary consolidation. Please explain in more detail how this model can be related to secondary consolidation.
Round 2
Reviewer 1 Report
Thank you for addressing all queries and suggestions. No additional comments.